# Sentiment Analysis on Twitter: Role of Healthcare Professionals in the Global Conversation during the AstraZeneca Vaccine Suspension

**DOI:** 10.3390/ijerph20032225

**Published:** 2023-01-26

**Authors:** Carlos Ruiz-Núñez, Ivan Herrera-Peco, Silvia María Campos-Soler, Álvaro Carmona-Pestaña, Elvira Benítez de Gracia, Juan José Peña Deudero, Andrés Ignacio García-Notario

**Affiliations:** 1Program in Biomedicine, Translational Research, and New Health Technologies, School of Medicine, University of Malaga, Blvr. Louis Pasteur, 29010 Malaga, Spain; 2Faculty of Health Sciences, Universidad Alfonso X el Sabio, Avda Universidad, 1, Villanueva de la Cañada, 28691 Madrid, Spain; 3Faculty of Medicine, Universidad Alfonso X el Sabio, Avda Universidad, 1, Villanueva de la Cañada, 28691 Madrid, Spain

**Keywords:** communication, healthcare professionals, public health, social media, Twitter, vaccines

## Abstract

The vaccines against COVID-19 arrived in Spain at the end of 2020 along with vaccination campaigns which were not free of controversy. The debate was fueled by the adverse effects following the administration of the AstraZeneca-Oxford (AZ) vaccine in some European countries, eventually leading to its temporary suspension as a precautionary measure. In the present study, we analyze the healthcare professionals’ conversations, sentiment, polarity, and intensity on social media during two periods in 2021: the one closest to the suspension of the AZ vaccine and the same time frame 30 days later. We also analyzed whether there were differences between Spain and the rest of the world. Results: The negative sentiment ratio was higher (U = 87; *p* = 0.048) in Spain in March (Med = 0.396), as well as the daily intensity (U = 86; *p* = 0.044; Med = 0.440). The opposite happened with polarity (U = 86; *p* = 0.044), which was higher in the rest of the world (Med = −0.264). Conclusions: There was a general increase in messages and interactions between March and April. In Spain, there was a higher incidence of negative messages and intensity compared to the rest of the world during the March period that disappeared in April. Finally, it was found that the dissemination of messages linked to negative emotions towards vaccines against COVID-19 from healthcare professionals contributed to a negative approach to primary prevention campaigns in the middle of the pandemic.

## 1. Introduction

The main tools to combat the spread of COVID-19 have been mass vaccination, social isolation measures, the use of masks, and hand disinfection [1]. The first vaccines, produced by Pfizer-BioNTech and Moderna, were distributed in the second half of 2020, following approval by the U.S. Food and Drug Administration (FDA) and the European Medicines Agency (EMA) [2,3]. In Europe there were joint purchases of vaccination batches promoted by the European Union [3].

During this period there were numerous campaigns against vaccination, on the grounds of not having been tested or not fully knowing their side effects. Some deficiencies in previous trials did not help either, such as the one reported by AstraZeneca and Oxford University due to a dosing error during the trials of their vaccine, which generated mistrust in the population [4]. The World Health Organization (WHO) itself considers vaccine hesitancy as one of the top 10 threats to global health [5].

During the year 2021, adverse effects caused by different batches of vaccines were reported in various countries. Thus, cases of thrombi were described after the administration of the AstraZeneca-Oxford vaccine [6] in Norway, Denmark, and Italy, or with Janssen vaccines in the United States [7]. This led to a temporary suspension of the vaccination campaign, with AstraZeneca vaccines in some European countries [4,8] including Spain in April 2021 [7] and with Janssen vaccines in the USA [9]. Suspension was eventually confirmed by the EMA as a precautionary measure in Europe [10]. In all cases, vaccine use was resumed days later, and the EMA itself concluded that the benefits of the vaccines outweighed the risk of side effects [11].

In a time like the COVID-19 pandemic, where the need for information was very high [12], social media, represented by the Internet in general and especially by social networks such as Twitter, Facebook, Instagram, TikTok, or YouTube [13,14,15], played a very important role for a part of the population. In addition, its relevance was boosted by the social restrictions derived from the preventive social isolation against COVID-19. Aside from the traditional use as a source of entertainment, we have witnessed how social media expanded into other more sophisticated roles, such as education or analysis of users’ sentiment [16].

Moreover, social media represents a relevant tool for looking at public health, due to its simplicity and low cost [17], and can also help in monitoring the population’s response to health problems [18,19]. However, it is important to highlight that social media has played an essential part in the dissemination of disinformation and fake news [20], having contributed to undermining the credibility of agencies and institutions on issues such as the vaccination campaigns, which were performed as a public health measure against COVID-19. Cases of adverse effects or reactions have been found in almost all vaccines marketed [21,22]. However, social networks and opinion websites charged against vaccination campaigns, focusing on the AstraZeneca vaccine [23,24]. This led to the implementation of filters on the main social media platforms to avoid the dissemination of misinformation [25] that could feed false theories.

Emotions and feelings come from perceptions and mental elements that involve internal and external processes [26], playing a key role in human relationships, for they are one of the pillars of social activity [27,28]. These interactions among people, something that also takes place on social media, generate inputs that may influence and modulate the behavior of other individuals receiving those messages, feelings, and emotions. [26].

The use of social media as a source of information for researchers experienced great relevance in recent years, including the pandemic period, with special emphasis on the analysis of feelings and emotions [29,30,31] not only based on texts, but also on emotional analysis of photographs and videos [32]. Although the analysis of the healthcare professional’s role in the dissemination of emotions and sentiment associated with vaccination cannot be found specifically in the literature, it has been studied consistently at a general level in mainstream users of social networks. Some studies conducted on Twitter in the U.S. highlighted positive emotions and sentiment towards vaccination for COVID-19 [20,33]. However, there are also studies that suggest that negative emotions and sentiment towards vaccination are in the majority, mostly when analyzing conversations not focused on vaccination campaigns [34].

The WHO states that, in order to manage an infodemic—an epidemic of information that may be false or untrue in digital and physical environments—and reduce its impact on health behaviors during health emergencies, there should be promoted deeper risk understanding and enhanced advice by health experts [35], among other measures. The group of experts must include healthcare professionals, who are expected to have superior knowledge based on training and experience. These skills should be leveraged to improve health literacy in society, communicating their knowledge through truthful, accessible, understandable, and useful messages. They must reach all individuals using any communication channel [36,37,38], including social networks.

The importance of social media for the management of an infodemic is indisputable, and healthcare professionals should take advantage of them for health literacy purposes. Our main goal in the present study was to know whether health professionals were affected by the surrounding news in their opinions on Twitter and how they convey these sentiments publicly.

Thus, the lessons learned during the pandemic can be transferred to the management of public health communication, as recommended by the WHO, including health professionals as educators in public health matters, such as vaccination campaigns, thanks to their training and professional ascendancy over the community, taking the example of what was learned during the pandemic.

In this context, the main objectives of the present study were (i) to analyze tweets and interactions of healthcare professionals on Twitter during the suspension of vaccination with AstraZeneca’s vaccine and its resumption in order to (ii) understand the behavior of feelings and their polarity and intensity, shown in Twitter messages created by healthcare professionals.

## 2. Materials and Methods

### 2.1. Study Design and Ethics

An observational, retrospective, and time-limited study was carried out, in which activity on the social network Twitter was analyzed, mining messages according to a previously determined hashtag.

This research focuses on messages posted by users on Twitter, measuring descriptive data and analyzing the sentiment of conversations on this social platform. The user’s profile information is public, as well as their publications, where the user configures their level of privacy so that the amount of information they want to publish to those who may not have any direct relationship with them is in their own hands. In this way, the visible data are those that the user themself wants to share and make public, which is why the approval of an Ethics Committee is not necessary for this research. However, any data that allowed unequivocal identification, either directly or inversely, were anonymized, contributing to good research practices in social networks [19]. The researchers are committed to protecting the confidentiality, integrity, and availability of the data.

### 2.2. Data Collection

In this study we approached the social network Twitter to explore, first globally and subsequently segmenting Spain versus the rest of the world, the conversations related to the suspension of the *AstraZeneca* vaccine, whenever profiles identified as healthcare professionals participated.

We focused on two periods of 17 days each, the first starting on 7 March 2021 and ending on 23 March of the same year, coinciding with the date on which the Austrian authorities announced the temporary suspension of vaccination with a batch of AstraZeneca’s vaccine, subsequently causing a cascading movement in Europe during the following days. The analysis was replicated during a second period from 7 April until 23 April 2021.

For the detection of keywords—or hashtags—to search for on Twitter, we used the Google Trends tool, finding that the most significant was #AstraZeneca.

Data mining was carried out with the academictwitteR library [20], connecting to Twitter, through Twitter’s API 2.0 application interface, with the free software R (https://www.r-project.org/) and its graphical interface RStudio, now called Posit. We entered the search parameters, which matched the inclusion criteria and the hashtag selected as representative of the study. The dataset obtained was downloaded in xlsx format for further manipulation (Figure 1).

For the study of sentiment, a data cleaning task was previously performed by removing symbols, emojis, hyperlinks, and empty words, in addition to splitting sentences into smaller units and converting text to lowercase.

All original tweets containing the keyword were included, along with the interactions that were prompted afterwards.

### 2.3. Variables for Analysis

In order to homogenize the data and allow comparison, we created a ratio between the messages associated to a specific sentiment and the total number of daily messages, which varies on a scale between 0 and 1. It resulted in two variables called daily ratio of positive sentiment and daily ratio of negative sentiment. More specifically, they were calculated by taking the number of messages with positive or negative sentiment divided by the total number of messages per day.

From these ratios we calculated two new variables, called polarity and intensity. The daily sentiment polarity was measured as the difference between the daily ratio of positive sentiment and the daily ratio of negative sentiment and can be interpreted according to two parameters: the sign indicates whether it is positive or negative, and the scale between 0 and 1 informs us on the strength of the daily conversation, once all vectors that converge have been compensated.

The last variable is the daily intensity of messages, including some kind of sentiment calculated by adding the daily ratio of positive sentiment and the daily ratio of negative sentiment. It indicates for us the magnitude of messages with sentiment in the daily conversation, on a scale between 0 and 1.

Regarding both polarity and intensity, the results can be reflected as percentages.

### 2.4. Data Analysis

In this study, an analysis was conducted on the sentiment of messages in conversations during the selected periods. It was possible to differentiate between emotions and sentiment, since the latter are more of an exploratory nature and break down into three levels (negative, neutral, and positive) based on the language of the tweets. The analysis of emotions refers to deeper levels, such as fear, sadness, anger, joy, or disgust, and it is considered a subsequent level of analysis, in greater depth than sentiment. In this study we always refer to sentiment analysis, using the Syuzhet library after adapting the data. For further research, a more complete analysis of emotions would be of interest.

Statistical analysis was performed with RStudio, IDE of the R language, and the corresponding libraries. We analyzed descriptive data, such as position and dispersion measures, and inferential statistics, with comparison of groups using Mann–Whitney U and setting the level of statistical significance at *p* < 0.05. The Shapiro–Wilk and Levene tests were used to verify the assumptions of normality and homoscedasticity, respectively.

## 3. Results

### 3.1. Descriptive Analysis

A total of 4,409,923 tweets related to the selected hashtag were collected worldwide during the aforementioned March period (Table 1), with 365,767 publications being posted from Spain (8.29%). In this same period, 44,710,821 global interactions were reached, with 1,288,171 (2.88%) corresponding to Spain with an average of 4,613,960.96 (Table 1).

During the April period, a total of 28,734 tweets were found worldwide, with 9716 (33.81%) coming from Spain (Table 2). Worldwide interactions reached 315,361, of which 113,058 (35.85%) came from Spanish accounts.

Adding the two periods, the studied hashtag generated worldwide 4,438,657 tweets and 45,026,182 interactions, which indicates that each message triggered more than 10 interactions (10.11) among the global Twitter audience. When comparing both periods, we can see that the volume of messages during the suspension of the vaccine versus the later period is significantly higher (*p* < 0.01), and a similar situation happened with interactions (*p* < 0.01).

In Spain, the total number of tweets was 375,483, generating 1,401,229 interactions. These figures yield a lower number of interactions per tweet compared to worldwide, at almost four per message (3.73). When analyzing the level of interactions generated by the messages, it was found that it was higher in the period of suspension of the AZ vaccine (*p* < 0.05), a pattern we also found in the generation of messages (*p* < 0.05).

### 3.2. Sentiment, Polarity, and Intensity

Messages with positive sentiments were analyzed. They amounted worldwide to 310,408 messages during the March period, of which 21,906 (7.05%) belonged to Spain. During April there were a total of 3190 global messages with positive sentiment, from which 32.88% (*n* = 1.049) belonged to Spain (Table 3).

We used the same procedure for the analysis of messages with negative feelings, finding worldwide within the period of March 1,582,804 messages, of which 149,655 (9.45%) corresponded to Spain. In the period of April, they decreased to 9978 in the world and to 3555 (35.62%) in Spain.

In global terms, we found 1,592,782 messages with negative feelings, split into 153,210 for Spain and 1,439,572 for the rest of the world.

We calculated the daily ratio of messages with positive sentiment, which had an average of 0.077 for the rest of the world (world without Spain) during March, while in Spain it dropped to 0.070. For the same period, the average daily ratio of negative sentiment messages for the rest of the world was 0.340 and in Spain it roses up to 0.389. Focusing on the April period, the daily ratio of positive messages in the rest of the world was 0.118, while in Spain was 0.094, and in both cases higher than the period previously studied. For messages with negative sentiment, the ratio in this period in the rest of the world was 0.323, lower than the previous period, likewise in Spain with 0.367 (Figure 2).

During the period of March, we can observe a polarity of negative sign in Spain and in the rest of the world, −0.319 against −0.264, respectively. Studying the April period, we found that the polarity is −0.273 for Spain and −0.216 for the rest of the world.

We found an intensity in Spain of 0.460 and 0.417 in the rest of the world. During April, the average in Spain was 0.461, very similar to the previous period, and 0.441 in the rest of the world, slightly higher compared to the March period (Figure 3).

### 3.3. Intragroup Analysis

We analyzed the behavior of messages generated between both groups, Spain and the rest of the world, within the two periods. During April, no significant statistical differences were observed.

However, during March, we found statistically significant differences in the daily ratio of negative sentiment (U = 87; *p* = 0.048), which proved higher in Spain (Mdn = 0.396) than in the rest of the world (Mdn = 0.342). The same situation was found in the daily intensity of messages (U = 86; *p* = 0.044), higher in Spain (Mdn = 0.440) than in the rest of the world (Mdn = 0.404). In the case of daily polarity the situation is the opposite, and we did find statistically significant differences (U = 86; *p* = 0.044), being higher in the rest of the world (Mdn = −0.272) than in Spain (Mdn = −0.342). However, at this point we must stick to the interpretation of the variable.

### 3.4. Intergroup Analysis

We analyzed the behavior of messages between both periods at different categories: worldwide level, rest of the world, and Spain.

With regards to Spain, no statistically significant differences were observed between March and April in any of the variables studied.

However, during the March period, we found statistically significant differences in the daily ratio of positive sentiment, both worldwide (U = 74; *p* = 0.015) and for the rest of the world (U = 82; *p* = 0.031), and higher in April (Mdn = 0.103 worldwide and Mdn = 0.102 for the rest of the world) versus March (Mdn = 0.077 and Mdn = 0.072, respectively).

## 4. Discussion

The present study analyzes the role of healthcare professionals and their influence on social media, as their presence is increasingly greater [39,40] in conversations about vaccination against COVID-19. This can be one of the essential aspects of prevention against the disease [41]. Specifically, one of most interesting areas was to study social media as a source of disinformation [42] during the pandemic in the period of March 2021, coinciding with the temporary suspensions of vaccination with the AstraZeneca vaccine [4,6,7,8,9].

In response to the first aim of this study, it is important to point out that differences were found both in the generation of messages and in the interactions produced. Globally and in Spain, there were significant differences in both the number of messages and interactions between the period of the precautionary suspension of vaccination and the subsequent one, when it was established that there were no risks and vaccination could be resumed [10]. It should be noted that, in the period corresponding to April, a smaller number of messages and interactions were observed, although far more present in Spain than in the rest of the world. The cause may lie in a late incorporation of Spain to social media conversations due to a belated suspension of the vaccine.

Based on these data, it can be deduced that healthcare professionals actively participated in Twitter commenting on the situation regarding the COVID-19 vaccines. However, the role of these professionals seems to be more focused on comments from a non-professional nature, in accord with the literature which indicates that healthcare professionals mostly use social networks with a personal approach [43,44].

The level of interactions generated by healthcare professionals in the conversations is high, a fact that may be related to the authority that the population confers to their opinions on health-related topics [45]. It also may be influenced by the fact that claims could be backed up with reliable sources of information [46] and communicated in a way that is suitable for the public to understand [47]. These are key elements for standard users of Twitter to be able to search, read, and share messages posted by healthcare professionals.

Finally, and in reply to the second aim asked initially when analyzing the predominant sentiments in the conversation during both periods, we observed that the same pattern existed for the daily ratio of both positive and negative messages: a decrease in April with respect to March. More specifically, the predominance of positive feelings was greater in the conversation of the category “rest of the world”, a pattern that has been already detected in previous analyses of COVID-19 [48] and in vaccination intention [49].

However, in Spain, we observed that the predominant sentiment in the conversations was negative, a conclusion consistent with several previous studies, although in this latter case, studies were carried out on the general population and not specifically on healthcare professionals [50,51]. The difference of emotional response on social networks between the two geographical groups may be due to many factors, yet some authors propose that it can be attributed to mainly two: ethnic and cultural aspects specific to each country [51].

Associated with analysis of the polarity and of sentiments in the conversation—something that is increasingly common on social networks such as Twitter—[52] showed that in both groups the sign was negative. Although Spain has a much higher value than the rest of the world, the fact that both have the same “direction” indicates that healthcare professionals have shown very significant negative sentiment, over 30%, being higher during the period of March than it is in April.

Although there is literature affirming the positive role of social media in global vaccination campaigns [49], we cannot ignore our findings. The main prescribers of truthful scientific information, healthcare professionals, have been transmitting a significant number of messages with negative sentiments and have influenced the polarity of the conversation. Within the global conversation, the most belligerent were in Spain.

The general sentiment detected, together with the polarity of the conversation in Spain, negative in both cases, can be associated with psychological processes that affected the population—since healthcare professionals are part of it. Feelings such as anxiety, stress, confusion, and even frustration might have appeared with regards to the information received about the COVID-19 pandemic [40,41,53] and influence the sentiments of the messages. Nevertheless, having sent negative messages regarding a public health measure such as vaccines [41] means that healthcare professionals themselves may have contributed to the dissemination of misinformation on health topics [53,54]. Moreover, it is essential to remember that many healthcare professionals, or users who identify themselves as such, use social networks for personal activity [43,44], arguably making use of freedom of speech to express their own opinions [45]. It is important to stress that, although social media is open to everyone, healthcare professionals should be very careful with the opinions stated on health issues, given the bias of authority existing in the society [45,46,47]. They must avoid expressing their opinions without having all the information and scientific evidence [47].

This study has several important limitations. The first is associated with the data collection, since the categorization as a healthcare professional is based on whether an individual defines themselves as such in the account description. In addition, their profiles must be publicly accessible. Secondly, it is possible that part of the conversation—tweets-—was not collected because users might have used different hashtags. To minimize this limitation, a thorough monitorization was conducted to find conversations on the same topic but taking place with other keywords. Another limitation may be related to a possible loss of messages due to being published in non-studied languages. Finally, another limitation to note is that the study focuses exclusively on Twitter, while there exist several other social networks, some of them focused on visual contents and the conversations taking place on them.

## 5. Conclusions

The results presented in this study provide information about the role of healthcare professionals in conversations about vaccines in the face of COVID-19. Specifically, it focused on how these professionals reacted to the news of a temporary suspension of a vaccine that had already been approved and administered to part of the population.

It is important to highlight that social media facilitates the spread of information, but also displays the sentiment and emotions of the messages’ authors. This aspect has not been addressed assiduously in the bibliography. Nevertheless, it is becoming increasingly important due to the uncertainty on health information and the risk that consumption of incorrect contents can pose to the population, ultimately affecting their level of health literacy.

In this situation, healthcare professionals may generate uncertainty in standard users of social networks, i.e., the population, contributing to the distrust towards validated scientific information that may come from health institutions.

Admittedly, it is very complex to take action to improve the healthcare professionals’ use of social media, but we do believe that training activities should be considered to help them improve their skills in research methodology, critical reading, and communication on platforms. We think that a training of this kind would contribute to prevent a careless use of the messages conveyed through social media, avoiding the dissemination of unverified information based solely on opinions, which is detrimental to primary prevention campaigns in health events such as the COVID-19 pandemic.

## Figures and Tables

**Figure 1 ijerph-20-02225-f001:**
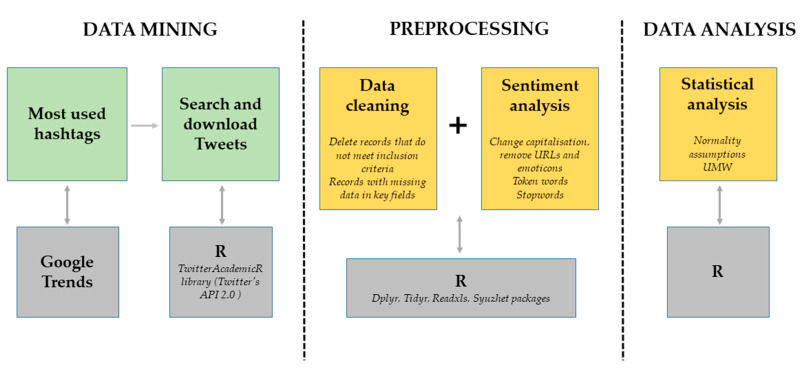
Data mining and analysis scheme.

**Figure 2 ijerph-20-02225-f002:**
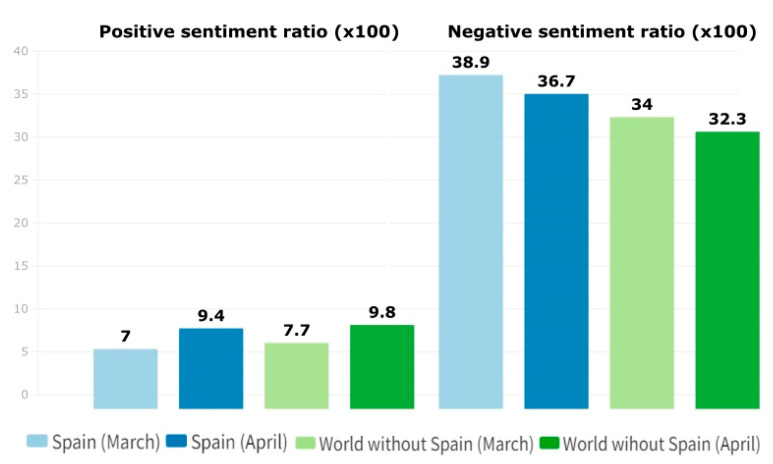
Average daily sentiment ratio.

**Figure 3 ijerph-20-02225-f003:**
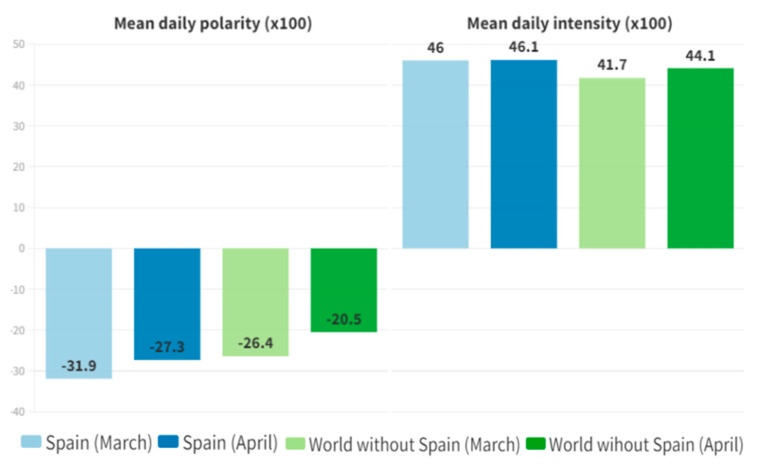
Daily polarity and intensity.

**Table 1 ijerph-20-02225-t001:** Messages on Twitter.

	**Worldwide 7 March 2021 to 23 March 2021**					
	** *n* **	**Mean**	**Median**	**95% CI**		**SD**
				**Lower**	**Top**	
**Tweets**	4,409,923	259,407.24	202,682.00	163,976.32	354,838.15	185,608.17
**Interactions**	4,4710,821	2,630,048.29	4,613,960.96	646,135.63	4,613,960.96	4,613,960.96
	**Worldwide 7 April 2021 to 23 April 2021**					
	** *n* **	**Mean**	**Median**	**95% CI**		**SD**
				**Lower**	**Top**	
**Tweets**	28,734	1690.24	1367.00	1230.23	2150.24	894.69
**Interactions**	315,361	18,550.65	8663.00	8990.28	28,111.01	18,594.41

**Table 2 ijerph-20-02225-t002:** Messages on Twitter, segmented between Spain and the rest of the world.

	**Spain**					**Rest of the World**				
	**7 March 2021 to 23 March 2021**									
	**Mean**	**Median**	**95% CI**		**SD**	**Mean**	**Median**	**95% CI**		**SD**
			**Lower**	**Top**				**Lower**	**Top**	
**Tweets**	21,515.71	15,821.00	13,141.97	29,889.44	16,286.49	237,891.53	186,861.00	147,696.68	328,086.38	175,424.30
**Interactions**	75,774.76	68,197.00	44,096.48	107,453.05	61,612.62	2,554,273.53	1,487,300.00	579,803.91	4,528,743.15	3,840,240.79
	**7 April 2021 to 23 April 2021**									
	**Mean**	**Median**	**95% CI**		**SD**	**Mean**	**Median**	**95% CI**		**SD**
			**Lower**	**Top**				**Lower**	**Top**	
**Tweets**	571.53	381.00	330.04	813.01	469.68	1118.71	1081.00	856.41	1381.01	510.16
**Interactions**	6650.47	1620.00	1636.26	11,664.68	9752.38	11,900.18	7558.00	5309.36	18,491.00	12,818.80

**Table 3 ijerph-20-02225-t003:** Sentiment, polarity, and intensity.

	**Spain**						**Rest of the World**					
	**7 March 2021 to 23 March 2021**											
	** *n* **	**Mean**	**Median**	**95% CI**		**SD**	** *n* **	**Mean**	**Median**	**95% CI**		**SD**
				**Lower**	**Top**					**Lower**	**Top**	
**Positive messages**	21,906	21,515.71	15,821.00	13,141.97	29,889.44	16,286.49	288,502	237,891.53	186,861.00	147,696.68	328,086.38	175,424.30
**Negative messages**	149,655	4,613,960.96	107,453.05	44,096.48	107,453.05	61,612.62	1,433,149	2,554,273.53	1,487,300.00	579,803.91	4,528,743.15	3,840,240.79
**Positive DSR**		0.070	0.054	0.045	0.096	0.050		0.077	0.072	0.063	0.090	0.027
**Negative DSR**		0.389	0.396	0.354	0.424	0.068		0.340	0.342	0.306	0.375	0.068
**Daily polarity**		−0.319	−0.342	−0.372	−0.266	0.104		−0.264	−0.272	−0.307	−0.220	0.085
**Daily intensity**		0.460	0.440	0.429	0.490	0.060		0.417	0.404	0.387	0.446	0.582
	**7 April 2021 to 23 April 2021**											
	** *n* **	**Mean**	**Median**	**95% CI**		**SD**	** *n* **	**Mean**	**Median**	**95% CI**		**SD**
				**Lower**	**Top**					**Lower**	**Top**	
**Positive messages**	1049	571.53	381.00	330.04	813.01	469.68	2141	1118.71	1081.00	856.41	1381.01	510.16
**Negative messages**	3555	6650.47	1620.00	1636.26	11,664.68	9752.38	6423	11,900.18	7558.00	5309.36	18,491.00	12,818.80
**Positive DSR**		0.094	0.082	0.060	0.128	0.066		0.118	0.102	0.081	0.154	0.071
**Negative DSR**		0.367	0.359	0.321	0.413	0.090		0.323	0.321	0.285	0.361	0.075
**Daily polarity**		−0.273	−0.287	−0.336	−0.211	0.121		−0.205	−0.216	−0.272	−0.139	0.129
**Daily intensity**		0.461	0.434	0.409	0.513	0.101		0.441	0.448	0.406	0.476	0.068

Where DSR = daily sentiment ratio.

## Data Availability

The data that support the findings of this study are available from the corresponding author upon reasonable request.

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
