# Peer review of "Sentiment Analysis on Twitter: Role of Healthcare Professionals in the Global Conversation during the AstraZeneca Vaccine Suspension"

_ijerph, 2023, doi:10.3390/ijerph20032225_

Round 1

Reviewer 1 Report

The manuscript "Sentiment analysis on Twitter: Role of healthcare professionals in the global conversation during the Astrazeneca vaccine suspension" aims to analyze the healthcare professionals’ conversations, sentiment, polarity, and intensity on social 19 media during two periods of 2021.

Some suggestions for the authors:

- The Abstract is too long, includes too many conclusions. Please shorten it and make it more concise (less introduction, results and conclusions). I suggest the removal of paragraphs from line 26 - 34;

- Highlight the original contribution of the paper to the literature in domain? Further, present how can these results improve a future similar event? Please elaborate on these two in the Introduction or Results;

- lines 110-117 are identical with an article published by the same authors in Journal Vaccines. Please adjust/change the paragraphs;

- in Q1, line 106 I would recommend to remove the "..." as it looks like it is progress, not finished;

-  in Materials and Methods, I highly suggest authors to draw a schema with the steps used in the research. This will assure replication of the study and supports future works;

- it is not clear how was obtained the dataset. please elaborate on this issue on the subsection 2.1 data collection;

- please justify the reason to include results from 17 days only (no more, no less) and the specific interval for the second period (April 07th until April 23rd, 2021);

- in Table 3, line 206, the authors declare to display "Sentiment, polarity, and intensity". I see they display the "Sentiment's polarity and intensity". Further, please state how the intensity was determined (the formula);

- line 298 " Feelings like anxiety, stress .. " are emotions and some lexicons analyse them besides polarity (please check Emolex lexicon developed by NRC) https://saifmohammad.com/WebPages/NRC-Emotion-Lexicon.htm 

-besides limits exposed, others are met: the lack of multilanguage approach, emotions not included and they would offer more explanations besides positive and negative classification;

Major issues:

- I consider the manuscript lacks a strong literature review. Besides a review of papers written in the "medical" component of the title, there should also be added at least a paragraphs for the investigation types that have been conducted in sentiment analysis component of the title. In this way, it should be mentioned that some authors analysed, emotions(10.1007/s13278-022-01000-9, 10.1016/j.ipm.2022.103151,10.32604/ijmhp.2022.022641, 10.1007/978-3-031-16203-9_33) not only polarity, photos versus video (Social media) posts (10.3390/su11164459), the inclusion or not of lexicons for multiple languages (10.11591/eei.v12i2.3914,10.1016/j.mex.2022.101960) and other aspect of sentiment analysis;

-the analysis is a bit outdated (we are in 2023 and the analysis is from 2021). but the authors can justify is still an actual topic as the Covid is still among us and the results can benefit in the case of another similar situation;

- it is not clear how the authors answered the research questions (RQ) Q1-Q3. Moreover, the research hypothesis (RH) are missing. RQ and RH should have been included in Materials and methods or In Results (the authors answered briefly to the Q in Conclusions). While the RQ and RH represents the backbone of a research, they must be included and given full attention while conducting the research.

I am confident the authors can solve the above aspects as they have a good material. The above suggestions are meant to be constructive for the sake of improvement of the current version of the manuscript. 

Author Response

We thank you for your comments and explain the queries raised, and hope that we have been able to answer them to your satisfaction.

Question 1: The Abstract is too long, includes too many conclusions. Please shorten it and make it more concise (less introduction, results and conclusions). I suggest the removal of paragraphs from line 26 - 34;

Reply to the reviewer : Thank you for your comment, it did indeed exceed the maximum limit recommended for the Abstract. We have modified it following your recommendations to adapt it to the recommendations of the journal.

Question 2: Highlight the original contribution of the paper to the literature in domain? Further, present how can these results improve a future similar event? Please elaborate on these two in the Introduction or Results;

Reply to the reviewer: Thank you for the tip. We have added it at the end of the Introduction section before the research questions.

Question 3: lines 110-117 are identical with an article published by the same authors in Journal Vaccines. Please adjust/change the paragraphs;

Reply to the reviewer: Thank you for your comment. We have redrafted the text.

Question 4: in Q1, line 106 I would recommend to remove the "..." as it looks like it is progress, not finished;

Reply to the reviewer: Thank you for your comment, we have modified it following your recommendations.

Question 5: in Materials and Methods, I highly suggest authors to draw a schema with the steps used in the research. This will assure replication of the study and supports future works;

Reply to the reviewer: Thank you for your comment, we Added figure 1.

Question 6: it is not clear how was obtained the dataset. please elaborate on this issue on the subsection 2.1 data collection;

Reply to the reviewer: Thanks for the recommendation, we have rewritten the data collection in subsection 2.1. Data collection.

Question 7: please justify the reason to include results from 17 days only (no more, no less) and the specific interval for the second period (April 07th until April 23rd, 2021);

Reply to the reviewer: The research started with AstraZeneca's vaccination suspensions, and monitoring publications. We observed a decrease in message flow after 15 days, although we adjusted the results to 17 days. We also wanted to compare whether the feelings persisted or decreased one month after the start of the study, so we performed the two periods indicated. The intermediate period without study serves to establish a cut-off and clarification of the data.

Question 8: in Table 3, line 206, the authors declare to display "Sentiment, polarity, and intensity". I see they display the "Sentiment's polarity and intensity". Further, please state how the intensity was determined (the formula).

Reply to the reviewer: The explanation can be found in subsection 2.2. Variables of analysis, where it appears "The last variable is the daily intensity of messages that include some kind of sentiment, calculated by adding the daily ratio of positive sentiment and the daily ratio of negative sentiment. It indicates the magnitude of messages with sentiment in the daily conversation, on a scale between 0 and 1".

Conversation intensity is therefore calculated as the sum of positive sentiment messages and negative sentiment messages, and provides an indicator of the total number of sentiment messages (intensity) present in conversations.

Question 9: line 298 " Feelings like anxiety, stress .. " are emotions and some lexicons analyse them besides polarity (please check Emolex lexicon developed by NRC) https://saifmohammad.com/WebPages/NRC-Emotion-Lexicon.htm 

Reply to the reviewer:  Indeed, anxiety and stress are emotions. This study deals with basic feelings, but we wanted to reflect that a more complete analysis of emotions can also be carried out using the NCR module found in the Syuzhet package we used. This is reflected in subsection 2.4. Data Analysis. We rewrite it to make it more understandable.

Question 10: besides limits exposed, others are met: the lack of multilanguage approach, emotions not included and they would offer more explanations besides positive and negative classification;

Reply to the reviewer:  Thank you for your comment, we have included possible language losses. As noted above, this study does not include sentiment analysis, although we have added sentiment analysis as a future research topic.

Regarding the study of emotions, it was not the purpose of this research, which is why it is not mentioned beyond a few bibliographical references.

Major issues:

Question 11: I consider the manuscript lacks a strong literature review. Besides a review of papers written in the "medical" component of the title, there should also be added at least a paragraphs for the investigation types that have been conducted in sentiment analysis component of the title. In this way, it should be mentioned that some authors analysed, emotions(10.1007/s13278-022-01000-9, 10.1016/j.ipm.2022.103151,10.32604/ijmhp.2022.022641, 10.1007/978-3-031-16203-9_33) not only polarity, photos versus video (Social media) posts (10.3390/su11164459), the inclusion or not of lexicons for multiple languages (10.11591/eei.v12i2.3914,10.1016/j.mex.2022.101960) and other aspect of sentiment analysis;

Reply to the reviewer:  We have included your recommendations as they improve the paper. Thank you very much for your comment.

Question 12: the analysis is a bit outdated (we are in 2023 and the analysis is from 2021). but the authors can justify is still an actual topic as the Covid is still among us and the results can benefit in the case of another similar situation;

Reply to the reviewer:  We agree with you that the dataset is from the year 2021, but we believe that the lessons it can provide for the future deserve to be analyzed. To find out how news, in this case, negative news, affects health professionals and whether it can help to improve communication in public health policies.

Question 13: it is not clear how the authors answered the research questions (RQ) Q1-Q3. Moreover, the research hypothesis (RH) are missing. RQ and RH should have been included in Materials and methods or In Results (the authors answered briefly to the Q in Conclusions). While the RQ and RH represents the backbone of a research, they must be included and given full attention while conducting the research.

Reply to the reviewer:  

Thank you very much for your advice, it was obviously confusing the way the objectives of the work were expressed.

To correct this mistake and facilitate the understanding of the article for future readers, we have modified all the text to which you refer in the introduction. Therefore, to facilitate the discussion of the objectives, we have proceeded to modify the discussion in such a way that it is in line with the objectives pursued in the development of this study.

Reviewer 2 Report

This is a well written article with a focus on the importance of social media messages from health care professional in influencing public perception on COVID-19 vaccine. 

However, I have the following comments:

There should be an explanation of how the author differentiated tweets from healthcare professionals from other tweets.

It would be helpful if the author can explain the process of annotation of the tweets into negative, positive and neutral

The article need to show some examples of the tweets. 

Was there a vectorization done for this study? If yes, the author should explain the methodology used.

Showing word cloud or other form of data visualization would be helpful.

Author Response

We thank you for your comments and explain the queries raised, and hope that we have been able to answer them to your satisfaction.

Question 1: There should be an explanation of how the author differentiated tweets from healthcare professionals from other tweets.

Reply to the reviewer: In section 2.1. Data collection indicate that the profiles have to be identified as health professionals, either in general or as a professional of a specific category (nurse, etc.). Let's take the Twitter user profile data and compare it with the different health professions, eliminating those that are not related.

Question 2: It would be helpful if the author can explain the process of annotation of the tweets into negative, positive, and neutral.

Reply to the reviewer: Before studying the sentiments we had to prepare the dataset with the downloaded messages and profiles (after filtering out those that were not health workers). First, we tokenize (segmentation of the text into sentences or words) the messages into words (token) and they are kept in the same order as in the text. We also convert uppercase to lowercase and remove punctuation marks, emoticons, numbers, and URLs, as well as confronting the tokens with stopwords, thereby creating a vector, in which we divide the string of characters into words and create a column with these words, with which the sentiment of each message is evaluated.

For sentiment evaluation, we use two arguments, first, a character vector (of words and phrases) and an opinion extraction method (syuzhet), which is the default for the package we use (syuzhet).

The syuzhet package comes with four sentiment dictionaries and provides a method for accessing the robust, but computationally expensive, sentiment extraction tool developed in the NLP group at Stanford.

Once the sentiment values of each tweet are determined, we categorize it as negative (-1), Neutral (0), or positive (1). With these values, we can then calculate variables such as average daily sentiment ratio, polarity, and intensity.

Question 3: The article needs to show some examples of the tweets.

Reply to the reviewer: Thank you very much for your opinion. We have not focused on individual tweets. The sentiment analysis is global and gives us a sentiment score per tweet. The idea of the paper is to check the overall sentiments of health workers, without individualizing. That is why we do not show individual tweets.

Question 4: Was there a vectorization done for this study? If yes, the author should explain the methodology used.

Reply to the reviewer: In the answer to question 2 you can find how we vectorized the messages.

Question 5: Showing word cloud or another form of data visualization would be helpful.

Reply to the reviewer: The study deals with the sentiments of the messages by performing a general analysis. After filtering and data preparation, information is removed, so a word cloud would not be meaningful. However, we appreciate your advice and for future manuscripts, we introduce this option.

We did perform it for hashtag choice but did not include it because we believed it was not meaningful for the paper.

Round 2

Reviewer 1 Report

The authors have addressed all of my suggestions and the new version of the manuscript is improved.

As such, I believe the manuscript can be considered for publishing within the Journal.